# Effect of Treadmill Exercise and Trans-Cinnamaldehyde against d-Galactose- and Aluminum Chloride-Induced Cognitive Dysfunction in Mice

**DOI:** 10.3390/brainsci10110793

**Published:** 2020-10-29

**Authors:** Jong-Sik Ryu, Ho-Youl Kang, Jong Kil Lee

**Affiliations:** 1Department of Physical Education, Kyungpook National University, 80, Daehak-ro, Buk-gu, Daegu 41566, Korea; jsryu1984@gmail.com (J.-S.R.); hokang@knu.ac.kr (H.-Y.K.); 2Department of Pharmacy, College of Pharmacy, Kyung Hee University, 26, Kyungheedae-ro, Dongdaemun-gu, Seoul 02447, Korea

**Keywords:** mild cognitive impairment, treadmill exercise, trans-cinnamaldehyde

## Abstract

Mild cognitive impairment (MCI) generally refers to impairment in cognition above that which accompanies the normal age-related cognitive decline and has attracted attention in recent years. Trans-cinnamaldehyde (TCA), which is isolated from cinnamon, has anti-inflammatory and antioxidant properties. Treadmill exercise also has diverse positive effects. The purpose of this study was to investigate the combination effects of TCA and treadmill exercise on learning and memory in a cognitive impairment mouse induced by a combination of d-galactose (d-gal) and aluminum chloride (AlCl_3_). We found that exercise and TCA attenuated cognitive impairment in mice with induced MCI. This effect was further increased by costimulation of exercise and TCA. To clarify the mechanisms of the positive effects of TCA and exercise, we analyzed the nuclear factor erythroid 2-related factor (Nrf2) and related signaling pathways. We found that TCA and exercise upregulated Nrf2, NAD(P)H dehydrogenase quinone 1 (NQO-1), heme oxygenase 1 (HO-1), and superoxide dismutase 1 (SOD-1); this suggests that TCA and exercise attenuate cognitive dysfunction by reducing oxidative stress. We also found that Nrf2-related signaling pathways, i.e., the AMP-activated protein kinase (AMPK)/Nrf2 and SIRT1/PGC-1a/Nrf2-ARE pathways, exerted antioxidant effects. Together, these results suggest that costimulation with TCA and exercise may be a therapeutic candidate for mild cognitive impairment.

## 1. Introduction

The brain is the chief main control center of the body, and it is very important to delay the aging of the brain for a healthy life [1]. Cognitive function decreases during the aging process of the brain, causing various problems such as impairment in information processing, attention, working, and long-term memory [2,3,4]. With the aging of societies, cognitive impairment is becoming an increasingly common disease, accompanied by social and economic problems [2]. Mild cognitive impairment (MCI) usually occurs before Alzheimer’s disease (AD), and it distinguished by the severity of cognitive decline that leads to functional impairment. For individuals with MCI, the risk of developing AD is much higher and in general, it is estimated that MCI transits from 5% to 10% per year into AD [5]. Therefore, it is important to explore effective therapeutics for memory dysfunction. 

Although the exact pathological mechanism underlying MCI or AD remains unclear, oxidative stress has been proposed as a critical factor in cognitive impairment [6]. Reactive oxygen species (ROS) have been reported to play a related role in the pathogenesis of several neurodegenerative diseases [7]. Lipid peroxidation, nuclear damage, and protein oxidation can occur due to overproduction of ROS, or decreased antioxidants in the brain [8]. Moreover, ROS are potent inducers of apoptosis which in turn contributes to the loss of neurons during memory impairment, thus finally affecting the normal functions of the brain [9].

Previous studies showed that the combined chronic administration of d-galactose (d-gal) and aluminum chloride (AlCl_3_) damages brain function, including learning and memory, and induces cognitive impairments [10,11,12]. d-gal treatment increases neurodegeneration by increasing the activity of acetylcholinesterase (AChE), ROS levels, and cognitive deficits and by inhibiting antioxidant enzymes [13]. Aluminum is an easily accumulated neurotoxic substance that blocks blood supply to the brain, causing cognitive impairment [14]. Therefore, co-treatment with AlCl_3_ and d-gal is used to mimic MCI and AD symptoms in rodents.

The benefits of exercise have been demonstrated in many human and experimental studies [15,16,17,18]. Regular exercise can be sufficient to repress the death of neuronal cells, not only animal models but also cognitive impairment patients and this indicates that physical activity or active life can prevent brain damage [19,20]. Physical activity improved cognitive functions by promoting neurogenesis, neurotransmission, and synaptic plasticity in previous animal studies [21,22,23].

Trans-cinnamaldehyde (TCA) is an aromatic aldehyde that is present in the bark extract of cinnamon. Previous studies have reported that cinnamon extracts have not only antioxidant, anti-sugar, and anti-inflammatory effects [24,25,26,27], but also neuroprotective activities in various diseases [28,29]. Considering the previous reports related to the positive effects of exercise and TCA, this study aimed to determine whether costimulation with exercise and TCA can ameliorate the cognitive impairments induced by d-gal and AlCl_3_ in mice.

## 2. Materials and Methods

### 2.1. Animals

Sixty male C57bl/6 mice (8 weeks of age, weighing 24–26 g at the beginning of the experiment) were purchased from Daehan Biolink Co., Ltd. All mice were housed in plastic containers under standard laboratory environments (temperature (23 °C ± 1 °C), humidity (60 °C ± 10 °C), 12-h light/dark cycle, and were given free access to food and water). All animal studies were performed per the “Principles of Laboratory Animal Care” (National Institutes of Health publication number 80–23, revised in 1996) and were approved by the Animal Care and Use Guidelines Committee of Kyung Hee University (approval number: KHUASP(SE)-19-041).

Mice were randomly divided into five experimental groups as follows:

Wild-type control group: control group treated with saline (WT, *n* = 12). 

d-gal + AlCl_3_ group: mild cognitive impairment group induced by d-gal + AlCl_3_ (MCI, *n* = 12).

d-gal + AlCl_3_ + TCA group: MCI group fed TCA (TCA, *n* = 12).

d-gal + AlCl_3_ + treadmill exercise group: MCI group that performed treadmill exercise (EX, *n* = 12).

d-gal + AlCl_3_ + TCA + treadmill exercise: MCI group that performed treadmill exercise and was fed TCA (TCA + EX, *n* = 12). 

### 2.2. Establishment of the Mild Cognitive Impairment Model

d-gal (G0750) and AlCl_3_ (294713) were purchased from Sigma Aldrich (St. Louis, MO, USA). The model of MCI was established by injecting d-gal (120 mg/kg/day) and AlCl_3_ (20 mg/kg/day) dissolved in saline (0.9% NaCl) intraperitoneally for 10 weeks, as described in previous studies [30,31,32]. The same volume of saline was injected in the control group during this period.

### 2.3. Trans-Cinnamaldehyde (TCA) Treatment

TCA (C80687) was purchased from Sigma Aldrich. TCA dissolved in saline (30 mg/kg/day) was injected intraperitoneally for 5 weeks. The same volume of saline was injected to the control group during the experimental period. The dose of TCA was as described previously [29].

### 2.4. Treadmill Exercise Protocol

The protocol of the treadmill exercise was adapted from previous studies [20,33,34]. An animal electric treadmill machine (JD-A-09, JEUNGDO Bio & Plant Co., Ltd., Seoul, Korea) was used in this study. Mice in the exercise groups were subjected to a familiarization period of treadmill exercise (10 min/day; the treadmill speed was increased from 5 m/min to 10 m/min daily step by step) of 1 week. After the adaptation period, mice performed the treadmill exercise. The details of the treadmill exercise protocol were as follows: 5 m/min for 5 min (warming-up), 8 m/min for 5 min, 12 m/min for 30 min, and 5 m/min for 5 min (cooling-down). Mice exercised five times a week between 5 p.m. and 8 p.m. for 9 weeks. During the treadmill excise, light electric stimulation was given to the end of the belt to induce the mice to run forward.

### 2.5. Y-Maze

The Y-maze methods was adapted from previous studies with a slight modification [35,36]. The Y-maze was 60 cm long and made of plastic, consisting of three passages in a triangular shape. The mice started in the center of the Y-maze and were given 8 min [37]. It was wiped with 70% ethanol and left for a sufficient time to avoid affecting the experiment during the trials. The number of correct alternations (as a percentage of all alternations) was taken as an index of the spatial working memory performance. The correct alternation was considered when the three passages were passed consecutively (i.e., ABC, BCA, or CAB but not ABA). The results were manually scored using the recorded videos.

### 2.6. Morris Water Maze (MWM)

The Morris water maze (MWM) was performed as previously described [38]. In brief, the mice swam in white colored water until arriving the hidden platform for maximum 60 s in one trial. One mouse had three trials in one day and trained for seven days. The latency times of each trial were checked. At day 8, the hidden platform was removed, and a probe task was performed. All trials and probe task were recorded, and the tracing data were analyzed using Toxtrac software (https://toxtrac.sourceforge.io).

### 2.7. Immunoblotting

Immunoblotting was performed as previously described [38]. In brief, the immunoblotting samples were made using the RIPA buffered lysates of mouse hippocampus. The samples (20 µg/10 μL) were separated by SDS-PAGE, followed by transfer to PVDF membranes (Merck, #IPVH00010, Darmstadt, Germany). The membranes were blocked for one-hour and incubated with primary antibodies at 4 °C overnight, followed by one-hour incubation with matched secondary antibodies. After antibody incubation, the membranes were developed with an enhanced chemiluminescence (ECL, Bio-Rad, 1705061) detection system and imaged using Fusion Solo S (Vilber, Collegien, France). The image analysis was conducted using Image J software (NIH, Bethesda, MD, USA). The antibodies were diluted in a 5% skim milk in TBS-T solution and all antibodies used in this study are listed in Table 1.

### 2.8. RNA Isolation and Quantitative Real-Time Polymerase Chain Reaction (qRT-PCR)

Quantitative real-time polymerase chain reaction (qRT-PCR) was performed as previously described [38]. In brief, total RNA of mouse hippocampus were extracted and converted to cDNA (3 μg/20 µL). The cDNA samples were amplified using SYBR Green (Enzynomics Daejeon, Republic of Korea) and the CFX Connect Real-Time PCR System (Bio-Rad Laboratories, Hercules, CA, USA). The primers used in this study listed in Table 2.

### 2.9. Statistical Analysis

All data were analyzed using SPSS version 25 (IBM corporation, Armonk, NY, USA). Statistical significance was defined as a *p* value less than 0.05. The date was shown as the mean ± standard error of the mean (SEM). Parametric tests such as ANOVA were used when the data satisfied the null hypothesis of the Levene’s test. Tukey’s post hoc test was performed if the *p* value was < 0.05 in one-way analysis of variance (ANOVA). For the non-parametric test, the Kruskal–Wallis test followed by Dunn’s post hoc multiple comparisons were used. GraphPad Prism 5.0 software (GraphPad software Inc., San Diego, CA, USA) was used to draw all graphs.

## 3. Results

### 3.1. TCA and Treadmill Exercise Improved Cognitive Function in Mice with a d-gal- and AlCl_3_-Induced Cognitive Deficiencies

To clarify whether TCA and treadmill exercise improve cognitive performance, we used the d-gal- and AlCl_3_-induced mouse model of cognitive deficiency (Figure 1A). During the experimental period, the body weight did not differ among the groups (Figure 1B). 

Spatial memory and working memory performance of mice were measured through Y-maze and MWM task. First, we conducted the Y-maze test to see how TCA and exercise treatments affect working memory performance in mice. In the case of the Y-maze, the MCI group showed significantly lower alternations than did the WT group, but this impairment was recovered in the TCA, EX, and TCA + EX groups (F = 8.765) (Figure 2A). The MWM task was used to assess spatial learning and memory by determining the time to find and reach the hidden platform. In the MWM task, the MCI group exhibited markedly impaired learning and memory compared with the WT group. Similar to the Y-maze results, the EX and TCA + EX groups exhibited recovery of memory compared with the MCI group (Figure 2B). Although there was no significant difference in the escape latency times between the MCI and TCA groups, the latter group exhibited an improvement tendency regarding memory (F = 4.771) (Figure 2B). The platform was removed after 7 consecutive days, and the mice were given 60 s to find the missing platform during the probe trial. The time spent in the target quadrant was significantly shorter in the MCI group compared with the WT, EX, and TCA + EX groups (Figure 2C). The probe task showed that there was no significant difference among the groups regarding the swimming speed and the total distance traveled (Figure 2D,E). Taken together, our results suggest that TCA or EX ameliorate memory dysfunction in mice with d-gal- and AlCl_3_-induced MCI, and that costimulation with exercise and TCA further increases memory function.

### 3.2. Effects of TCA and Exercise on the Nrf2 Signaling Pathway in the Brain of d-gal- and AlCl_3_-Treated Mice

Previous studies have shown that TCA and EX are effective antioxidants [25,39,40]. Antioxidant proteins are regulated by the expression of Nrf2, which protects against the oxidative damage triggered by inflammation and injury. Moreover, previous studies reported that cognitive function is associated with the Nrf2 signaling pathway [41,42]. Therefore, we examined whether the improved cognitive performance observed in MCI mice after the administration of TCA and treadmill exercise was related to the Nrf2 signaling pathway. As shown in Figure 3A, the MCI group exhibited decreased Nrf2 levels compared with the WT group. Levels of Nrf2 were increased in the TCA and EX groups, although this difference did not reach statistical significance (Figure 3A and Appendix A). The TCA+EX group exhibited a dramatically increased Nrf2 level compared with the MCI group (F = 3.377) (Figure 3A and Appendix A), indicating that costimulation with TCA and treadmill exercise might restore cognition through the Nrf2 signaling pathway. To clarify the antioxidant effect induced by Nrf2 signals after treadmill exercise and TCA, the hippocampal levels of the NQO1, HO-1, and SOD-1 proteins, which are representative antioxidant enzymes, were evaluated using immunoblotting. The antioxidant proteins were downregulated in the MCI group compared with the WT group, but were significantly elevated in the TCA and EX groups. Similar to the Nrf2 expression pattern, costimulation further upregulated NQO1 (F = 3.880), HO-1 (F = 3.727) and SOD-1 (F = 6.501) (Figure 3B,D and Appendix A). These results suggest that the memory improvement afforded by TCA and EX might be related to an antioxidant effect regulated by the Nrf2 signaling pathway.

### 3.3. TCA and Treadmill Exercise Triggered LKB1/AMPK and SIRT1/PGC1-α Expression in d-gal- and AlCl_3_-Treated Mice

AMPK is heterotrimeric serine/threonine enzyme which consists of a catalytic subunit and two regulatory subunits, that function monitoring sensor of the cellular energy status. In addition, AMPK diminishes oxidative stress and other potentially adverse cellular events [43]. To examine whether the antioxidant effect of TCA and EX was induced by AMPK activation, we used immunoblotting analyses to measure the changes in cerebral AMPK activity. We found that the activity of AMPK was decreased in the MCI group. In contrast, the activity of AMPK was increased in the EX group and, furthermore, there was a marked increase in the costimulation group (F = 4.144) (Figure 4A and Appendix A). An upstream kinase, serine threonine liver kinase B1 (LKB1), was also increased in the TCA, EX, and costimulation groups compared with the MCI group (F = 4.819) (Figure 4B). Next, we examined the silent information regulator T1 (SIRT1), which is a protein that coexists with AMPK [44]. Similar to AMPK activation, SIRT1 was also increased in the TCA, EX, and costimulation groups (F = 5.417) (Figure 4C). The mitochondrial peroxisome proliferator-activated receptor gamma coactivator-1alpha (PGC-1α) is the critical regulator of the activation of Nrf2 [45,46]. Both AMPK and SIRT1 regulate each other and they share common target molecules, for example PGC-1 α [47]. Therefore, we analyzed PGC-1α expression. As expected, PGC-1α was also upregulated in the TCA, EX, and TCA + EX groups compared with the MCI group (F = 5.065) (Figure 4D and Appendix A). Collectively, our data indicate that TCA and EX regulate the SIRT1/LKB1AMPK signaling pathway.

## 4. Discussion

The aim of this study was to examine the effect of TCA and treadmill exercise on behavioral dysfunction and neurological deficiencies of a mouse model with induced cognitive impairment. We found that TCA treatment and exercise activated the SIRT1/LKB1/AMPK signaling mechanism-mediated Nrf2 pathway, to exert antioxidant effects. Therefore, the combination of TCA and exercise might be an effective therapy for aging-associated diseases, including MCI. Notably, this was the first description of the co-stimulatory effect of TCA and exercise on the learning and memory impairment of the d-gal and AlCl_3_-induced mouse model.

The vast majority of cases of cognitive impairment are sporadic; however, the causes of these cases remain largely unknown. Therefore, the choice of animal model depends on the experimental hypothesis. Oxidative stress promotes neurotoxicity via inflammatory events, such as the production of ROS; therefore, oxidative stress is widely recognized to be very important in CNS physiology and pathophysiology [48]. Overproduction of ROS can cause protein oxidation, that acts as an important trigger of cognitive impairments and also lipid peroxidation, nuclear and mitochondrial DNA damage [8]. Because of these negative effects, oxidative stress has been consistently noted as a therapeutic target to treat neurodegenerative diseases [49]. In Our study, we chose a cognitive impairment mouse model induced by oxidative stress. d-gal treatment increases neurodegeneration by elevating ROS levels via the inhibition of antioxidant enzymes. Moreover, treatment with AlCl_3_ induces neurodegenerative symptoms similar to those of AD in rodents. Therefore, combination of d-gal- and AlCl_3_-treated mice were a suitable choice in this study to examine the effects of TCA and exercise on oxidative stress-induced cognitive impairment. 

TCA, a major component of cinnamon, is known to be effective in anti-inflammatory, antidiabetic, and antioxidant effects [24,25,26,27]. Exercise also improves health in various ways, including via anti-inflammatory and antioxidant actions and even by improving cognitive functions [21,50,51]. In particular, aerobic exercise has improved cognitive processes not only in animal studies but also in studies of older human subject with cognitive disabilities, even the elderly without cognitive impairment. [52,53]. Because of these effects, we hypothesized that TCA and exercise might protect against d-gal- and AlCl_3_-induced cognitive impairment in mice. We showed that TCA and exercise yielded an improvement in cognitive dysfunction, and that costimulation using TCA combined with exercise further increased these effects.

In this study, we investigated whether combined treatment of nutrition (through TCA) and exercise (through treadmill exercise) might improve the learning and memory of mice induced with cognitive impairment. For this purpose, Y-maze and MWM task were used. Our results showed that TCA and treadmill exercise yielded an improvement in cognitive dysfunction, and that costimulation using TCA combined with exercise further increased these effects.

Nrf2 and its target antioxidants provide a defense system against oxidative stress. Nrf2 is pivotal in the regulation of the cellular redox role by controlling the expression of over 200 downstream genes encoding Phase II response enzymes during oxidative challenge [54]. When Nrf2 activated it translocates into the nucleus. And Nrf2 binds to antioxidants response elements (AREs) to activate the expression of target genes, including *NQO1*, *HO-1*, and *SOD-1* [55]. To determine the manner in which TCA and exercise improve cognitive function in the mouse model with d-gal and AlCl_3_-induced MCI, we analyzed the Nrf2 signaling pathway. We found that TCA and exercise significantly upregulated the Nrf2/ARE pathway, resulting in the elevation of *NQO1*, *HO-1*, and *SOD-1*. These results suggest that TCA and exercise improve memory function via antioxidant effects, and that costimulation with TCA and exercise further increased these effects.

Several signaling cascades, such as the mitogen-activated protein kinases and AMPK, regulate the nuclear activation of Nrf2. AMPK is known to be effective in various diseases, such as neurological disorders, cancer, diabetes, and cardiovascular disease [56,57,58]. Previous studies reported that AMPK also mediated enhancement of Nrf2 signaling [59]. We found that TCA and exercise increased AMPK activation. We also found that TCA and exercise upregulated LKB1, which is an upstream molecule of AMPK and a member (together with another 12 proteins) of the Ser/Thr kinase family, which is closely related to AMPK [60]. LKB1 is an important enzyme, the deacetylation or phosphorylation of which is crucial to the activation of AMPK. SIRT1 is expressed widely in mammalian cells and has been reported in many tissues, including the brain [61]. Several reports have suggested that SIRT1 and AMPK have common activators, actions, and target molecules [44,47,62]. Similar to previous works showing the partnership between SIRT1 and AMPK, we found that SIRT1 was also increased in the exercise and costimulation groups. PGC-1α, which is a key enzyme associated with increased expression of Nrf2, was a common target molecule regulated by AMPK and SIRT1. TCA and treadmill exercise had a significant effect on the expression of PGC-1α. All things taken together, our studies suggest that TCA and exercise activate the SIRT/LKB1/AMPK signaling mechanism and result in an Nrf2-mediated antioxidant effect.

## 5. Conclusions

In summary, cognitive improvement afforded by TCA and exercise in mice with MCI induced by d-gal and AlCl_3_ was demonstrated in this study. TCA and exercise yielded improvement of learning and memory through Nrf2-mediated antioxidant effects. Moreover, the positive effects of TCA and exercise were related with the SIRT/LKB1/AMPK signaling pathway. Although many aspects of the exact partnership between SIRT1 and AMPK require further study, our results indicate that TCA and exercise stimulate AMPK and SIRT1, suggesting a significant correlation between these two molecules. Based on our results, we propose that the combined treatment of nutrition (through TCA) and exercise (through treadmill exercise) can help diseases associated with cognitive deficiency.

## Figures and Tables

**Figure 1 brainsci-10-00793-f001:**
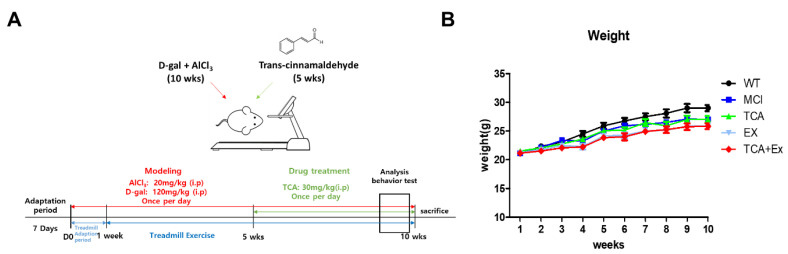
Experimental design and weight variation. (**A**) Schedule of the experiment. Eight-week-old C57Bl/6 mice were injected intraperitoneally with d-gal (120 mg/kg/day) and AlCl_3_ (20 mg/kg/day) for 10 weeks. Trans-cinnamaldehyde (TCA) (30 mg/kg/day) was injected intraperitoneally for 5 weeks. Treadmill exercise lasted 9 weeks and was initiated after 1 week of adaptation. Mice were euthanized and brain samples were collected for further analysis after the Morris water maze (MWM). (**B**) Ten weeks of d-gal and AlCl_3_ treatment induced cognitive impairment, and 5 weeks of TCA treatment and 9 weeks of treadmill exercise yielded no significant differences in weight among the groups. WT; wild-type control group, MCI; mild Cognitive impairment group, TCA; MCI group fed TCA, EX; MCI group that performed treadmill exercise. TCA+EX; MCI group that performed treadmill exercise and was fed TCA.

**Figure 2 brainsci-10-00793-f002:**
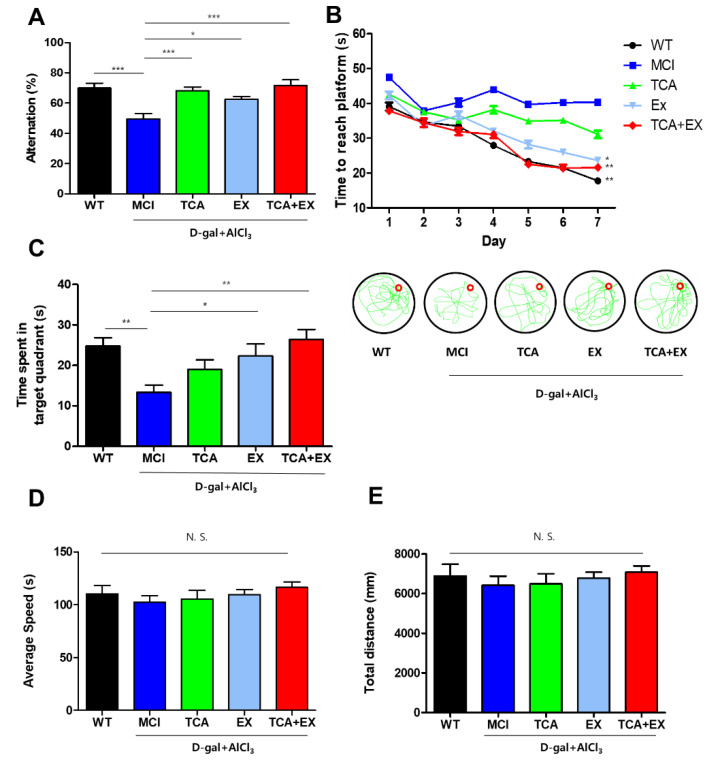
Exercise and TCA improved cognitive function in mice with d-gal- and AlCl_3_-induced cognitive deficiencies. (**A**) Spontaneous alternation scores in the Y-maze task. The occurrences of correct alternations are expressed as a percentage the total alternations (mean + standard error of the mean (SEM); *n* = 12 per group). (**B**) Spatial learning and memory of mice were analyzed by the time to reach platform for 7 days on the MWM. To analyze cognitive function, we measured the time required to reach the platform. (**C**) After 7 days, a probe task was performed. The amount of time that the mouse spent in the target quadrant was measured during 60 s probe test. To accurate analysis of memory, the average swim speed (**D**) and total distance traveled (**E**) were identified in the probe test. All results are expressed as the mean ± SEM; *n* = 12 per group). N.S.; no significant. * *p* < 0.05 compared with the MCI group; ** *p* < 0.01 compared with the MCI group; *** *p* < 0.001 compared with the MCI group. WT; wild-type control group, MCI; mild cognitive impairment group, TCA; MCI group fed TCA, EX; MCI group that performed treadmill exercise. TCA+EX; MCI group that performed treadmill exercise and was fed TCA.

**Figure 3 brainsci-10-00793-f003:**
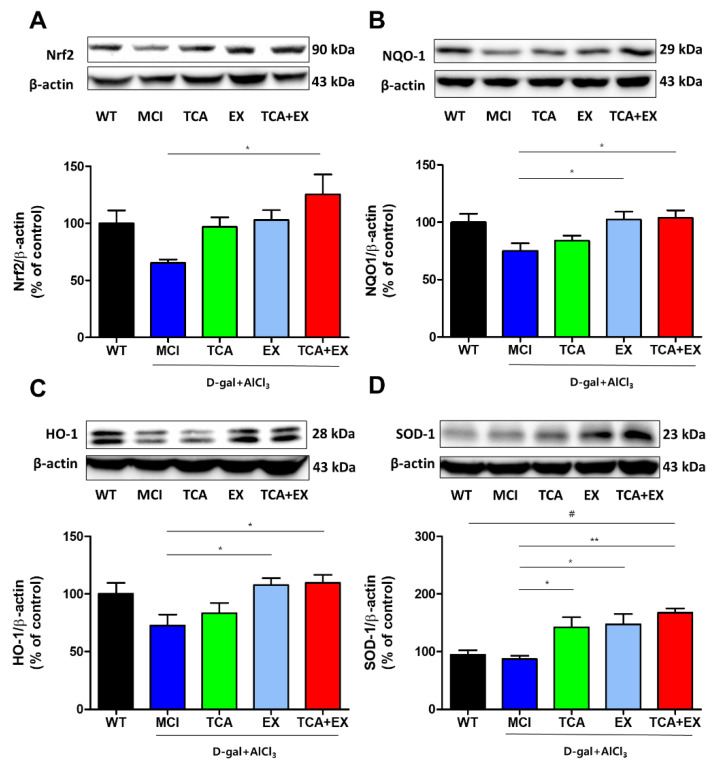
Effects of TCA and treadmill exercise on the Nrf2 signaling pathway in the brains of d-gal- and AlCl_3_-treated mice. (**A**–**D**) Representative immunoblot images and quantification of proteins related to the Nrf2 signaling pathway. Immunoblotting was carried out using antibodies against Nrf2 (**A**), NQO1 (**B**), HO-1 (**C**), and SOD-1 (**D**) on total protein lysates from the brain. All results are expressed as the mean ± SEM; *n* = 5–6 per group). * *p* < 0.05 compared with the MCI group; ** *p* < 0.01 compared with the MCI group; # *p* < 0.01 compared with the WT group. WT; wild-type control group, MCI; mild cognitive impairment group, TCA; MCI group fed TCA, EX; MCI group that performed treadmill exercise. TCA+EX; MCI group that performed treadmill exercise and was fed TCA.

**Figure 4 brainsci-10-00793-f004:**
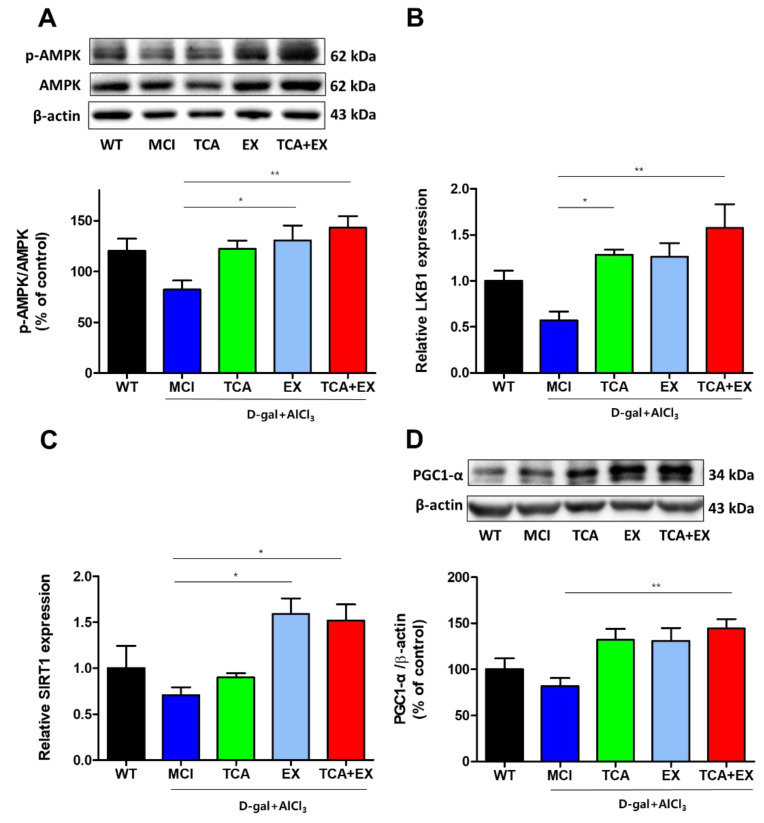
Effects of TCA and treadmill exercise on LKB1/AMPK and SIRT1/PGC-1α expression in d-gal- and AlCl_3_-treated mice. (**A**) Representative immunoblot images and quantification of the AMPK (**A**) and PGC-1α proteins (**D**). mRNA expression levels of LKB1 (**B**) and SIRT1 (**C**). All results are expressed as the mean ± SEM; *n* = 5–6 per group). * *p* < 0.05 compared with the MCI group; ** *p* < 0.01 compared with the MCI group. WT; wild-type control group, MCI; mild cognitive impairment group, TCA; MCI group fed TCA, EX; MCI group that performed treadmill exercise. TCA+EX; MCI group that performed treadmill exercise and was fed TCA.

**Table 1 brainsci-10-00793-t001:** Information of immunostaining antibodies used in this study.

Target	Host	Source	Catalog No.	RRID	Application
Nrf2	Rabbit	Cell signaling,	12721	AB_2715528	WB, 1:1000
NQO1	Rabbit	Cell signaling,	62262	AB_2799623	WB, 1:1000
HO-1	Rabbit	Cell signaling,	5853	AB_10835857	WB, 1:1000
SOD-1	Mouse	Santa Cruz Biotechnology,	271014	AB_10611197	WB, 1:1000
Phosphor AMPK alpha	Rabbit	Cell signaling,	2535	AB_331250	WB, 1:500
AMPK alpha	Rabbit	Cell signaling,	2532	AB_330331	WB, 1:500
PGC-1α	Mouse	Santa Cruz Biotechnology,	sc-518025	AB_2755043	WB, 1:1000
β-actin (HRP)	Mouse	Santa Cruz Biotechnology,	sc-47778 HRP	AB_2714189	WB/1:5000
Mouse IgG (HRP)	Goat	Santa Cruz Biotechnology	sc-2005	AB_631736	WB/1:5000
Rabbit IgG (HRP)	Goat	Santa Cruz Biotechnology	sc-2054	AB_631748	WB/1:5000

RRID; research resource identifiers, WB; western blot.

**Table 2 brainsci-10-00793-t002:** Information of quantitative real-time polymerase chain reaction (qRT-PCR) primers used in this study.

Gene Name	Forward Primer (5′→3′)	Reverse Primer (5′→3′)
LKB1	5-AGCTGCGCAGGATCCCCAAT-3′	5-TGGCACACAGGGAAGCGCTT-3′
SIRT1	5′-ACGCTGTGGCAGATTGTTATTA-3′	5′-TTGAAGAATGGTCTTGGGTCTT-3′
GAPDH	5′- TGAATACGGCTACAGCAACA-3′	5′-AGGCCCCTCCTGTTATTATG-3′

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
