# Peer review of "Effect of Treadmill Exercise and Trans-Cinnamaldehyde against d-Galactose- and Aluminum Chloride-Induced Cognitive Dysfunction in Mice"

_brainsci, 2020, doi:10.3390/brainsci10110793_

Round 1
Reviewer 1 Report
The manuscript is clearly written and the main goals are properly explained. The text only requires minor spell check.
In addition, I recommend the following minor revisions:
Add more recent references in the introduction, since there are many studies investigaing the benefits related to physical exercise and the consumption of natural extracts. In line 53, after "cognitive impairment" add the corresponding reference.
In the material and methods section:
Line 142: Add the company and cat# of the PVDF membranes used for the WB.
Line 142: Specify the solution used to dilute the primary antibodies for the WB.
Line 149: Provide details for the secondary antibodies used in the WB.
In figure 1:
Make more visible Fig.1A, in particular the writing in the upper part of the figure is difficult to read.
In the result section:
Please, provide the original images of the immunoblots in supplementary figures. In general the images appear to be manipulated too much with contrast and brightness.
The blot in figure 3D (SOD-1) should be changed since the WT band does not reflect what indicated in the text.
Lines 247-248: Specify that the phosphorylated form of AMPK is mainly upregulated upon treatment with TCA and EX.
Reviewer 2 Report
Ryu Brain Sci 2020
This is a worthwhile study on exercise and pharmacotherapy in a MCI model. The introduction gives a good idea on why the experiment was performed. More detail should be given in the methods section. What was the Y-maze made of, wood or steel? Did you clean the maze with water after every mouse trial or not, or did you use ethanol to clean? Was low-level light used? Was the video tracking software used in the Morris maze used as well in the Y-maze? If not, how were the mice followed? F values on ANOVA should be included in the results section, not only P values. The discussion is good but the conclusion on the behavioral results is too short (lines 291-293). What is mentioned on lines 194-196 should be mentioned again in the discussion. The figures are good. The references should be homogenized, for example references 29, 30, 40, and 47 are in capital letters whereas other references are not.
